# Subcellular tracking reveals the location of dimethylsulfoniopropionate in microalgae and visualises its uptake by marine bacteria

Jean-Baptiste Raina[1,2,3,4,5]*, Peta L Clode[6,7], Soshan Cheong[8], Jeremy Bougoure[6,9], Matt R Kilburn[6], Anthony Reeder[6], Sylvain Forêt[4,10†], Michael Stat[11], Victor Beltran[2], Peter Thomas-Hall[2], Dianne Tapiolas[2], Cherie M Motti[1,2], Bill Gong[8], Mathieu Pernice[3], Christopher E Marjo[8], Justin R Seymour[3], Bette L Willis[1,4,5], David G Bourne[2,5]

[1]AIMS@JCU, James Cook University, Townsville, Australia; [2]Australian Institute of Marine Science, Townsville, Australia; [3]Climate Change Cluster, University of Technology Sydney, Sydney, Australia; [4]ARC Centre of Excellence for Coral Reef Studies, James Cook University, Townsville, Australia; [5]College of Science and Engineering, James Cook University, Townsville, Australia; [6]The Centre for Microscopy Characterisation and Analysis, The University of Western Australia, Crawley, Australia; [7]Oceans Institute, The University of Western Australia, Crawley, Australia; [8]Mark Wainwright Analytical Centre, University of New South Wales, Kensington, Australia; [9]School of Earth and Environment, The University of Western Australia, Crawley, Australia; [10]Research School of Biology, Australian National University, Canberra, Australia; [11]Trace and Environmental DNA Laboratory, Department of Environment and Agriculture, Curtin University, Perth, Australia

*For correspondence: Jean-Baptiste.Raina@uts.edu.au

†Deceased

**Competing interests:** The authors declare that no competing interests exist.

**Abstract** Phytoplankton-bacteria interactions drive the surface ocean sulfur cycle and local climatic processes through the production and exchange of a key compound: dimethylsulfoniopropionate (DMSP). Despite their large-scale implications, these interactions remain unquantified at the cellular-scale. Here we use secondary-ion mass spectrometry to provide the first visualization of DMSP at sub-cellular levels, tracking the fate of a stable sulfur isotope ($^{34}$S) from its incorporation by microalgae as inorganic sulfate to its biosynthesis and exudation as DMSP, and finally its uptake and degradation by bacteria. Our results identify for the first time the storage locations of DMSP in microalgae, with high enrichments present in vacuoles, cytoplasm and chloroplasts. In addition, we quantify DMSP incorporation at the single-cell level, with DMSP-degrading bacteria containing seven times more $^{34}$S than the control strain. This study provides an unprecedented methodology to label, retain, and image small diffusible molecules, which can be transposable to other symbiotic systems.

## Introduction

Interactions between marine phytoplankton and bacteria constitute an important ecological linkage in the oceans (*Cole, 1982*), controlling chemical cycling and energy transfer to higher trophic levels (*Azam and Malfatti, 2007*; *Falkowski et al., 2008*). The cycling of sulfur, an essential element for living organisms, depends on the metabolic interactions between these two Kingdoms

**eLife digest** Sulfur is an essential element for many organisms and environmental processes. Every year, organisms including microalgae produce more than one billion tons of a sulfur-containing compound called DMSP. Some of this DMSP is released into seawater, where it acts as a key nutrient for microscopic organisms and as a foraging cue to attract fish. DMSP is also the precursor of a gas that helps to form clouds.

Despite DMSP's potential large-scale effects, it is still not clear what role it plays in the organisms that produce it, or how it is transferred from the microalgae that produce it to the bacteria that use it. It is thought that DMSP could potentially protect the cells from sudden changes in the amount of salt in the seawater (salinity) or from other damage, such as oxidative stress – a build-up of harmful chemicals inside cells.

In a controlled setting using artificial seawater, Raina et al. used high-resolution imaging and chemical analysis to track the journey of DMSP from microalgae to recipient bacteria. The results show that similar to land plants, algae store DMSP in the compartments that regulate cell pressure and photosynthesis. The presence of DMSP in these locations also supports its proposed role in protecting cells from changes in salinity or oxidative damage.

A future step will be to identify the genes involved in producing DMSP in microalgae. This knowledge could be used to create mutants that are either incapable of producing this molecule or that overproduce it. In combination with the high-resolution imaging techniques described here, this will allow researchers to fully understand the role that DMSP plays in these organisms.

(*Sievert et al., 2007*). A striking example is the production of the sulfur compound dimethylsulfonio-propionate (DMSP) by phytoplankton and its degradation by marine bacteria (and phytoplankton themselves) into the climate-active gas dimethylsulfide (DMS) (*Alcolombri et al., 2015*; *Ayers and Gras, 1991*; *Howard et al., 2006*; *Todd et al., 2007*). The subsequent release of DMS into the atmosphere contributes 90% of biogenic sulfur emissions and initiates the formation and growth of aerosols, thereby enhancing cloud formation and sunlight scattering (*Ayers and Gras, 1991*). This highlights how chemical interactions occurring between marine microorganisms across micrometre-scales can ultimately have large-scale impacts on climate (*Sievert et al., 2007*; *Simó, 2001*). However, direct measurements of these metabolic interactions, critical to the global sulfur cycling, have not previously been possible at the scale where they occur, the sub-cellular level.

In the surface ocean, the largest quantities of sulfur are present as dissolved sulfate, which constitutes the main sulfur source for phytoplankton (*Sievert et al., 2007*; *Stefels, 2000*). Most of the sulfur derived from sulfate uptake is converted by these organisms into sulfur-based amino acids, and a fraction is ultimately used to synthesise DMSP (*Stefels, 2000*) (*Figure 1*). Globally, more than a billion tons of DMSP are produced every year, which has been estimated to represent up to 10% of the amount of carbon fixed by phytoplankton (*Archer et al., 2001*; *Simó et al., 2002*). However, despite the central role played by DMSP in the marine sulfur cycle, a mechanistic understanding of the biochemistry at the heart of DMSP cycling is currently lacking. Previous studies in higher plants provided strong evidence that DMSP biosynthesis starts in the cytosol and ends in the chloroplast (*Trossat et al., 1996*, *1998*). However, DMSP biosynthesis occur through a different route in phytoplankton (*Stefels, 2000*), and we still do not know: (1) where this compound is produced and stored in phytoplankton cells; (2) what are its functions; and (3) how efficiently it is transferred from phytoplankton producers to bacterial degraders.

We used the dinoflagellate *Symbiodinium*, a taxon that includes some of the most prodigious DMSP producers on the planet (*Caruana and Malin, 2014*; *Saltzman and Cooper, 1989*). *Symbiodinium* cells can be free-living in the water column, but are primarily known for the endosymbiotic associations they form with tropical cnidarians that fuel the extremely high productivity of coral reef ecosystems (*Dubinsky, 1990*). Populations of reef-building corals are major DMSP production hotspots (*Broadbent et al., 2002*; *Raina et al., 2013*) and their contribution to the marine sulfur cycle is disproportionately large given their relatively restricted distributions (*Raina et al., 2013*; *Fischer and Jones, 2012*). In this ecosystem, DMSP constitutes an important source of carbon and

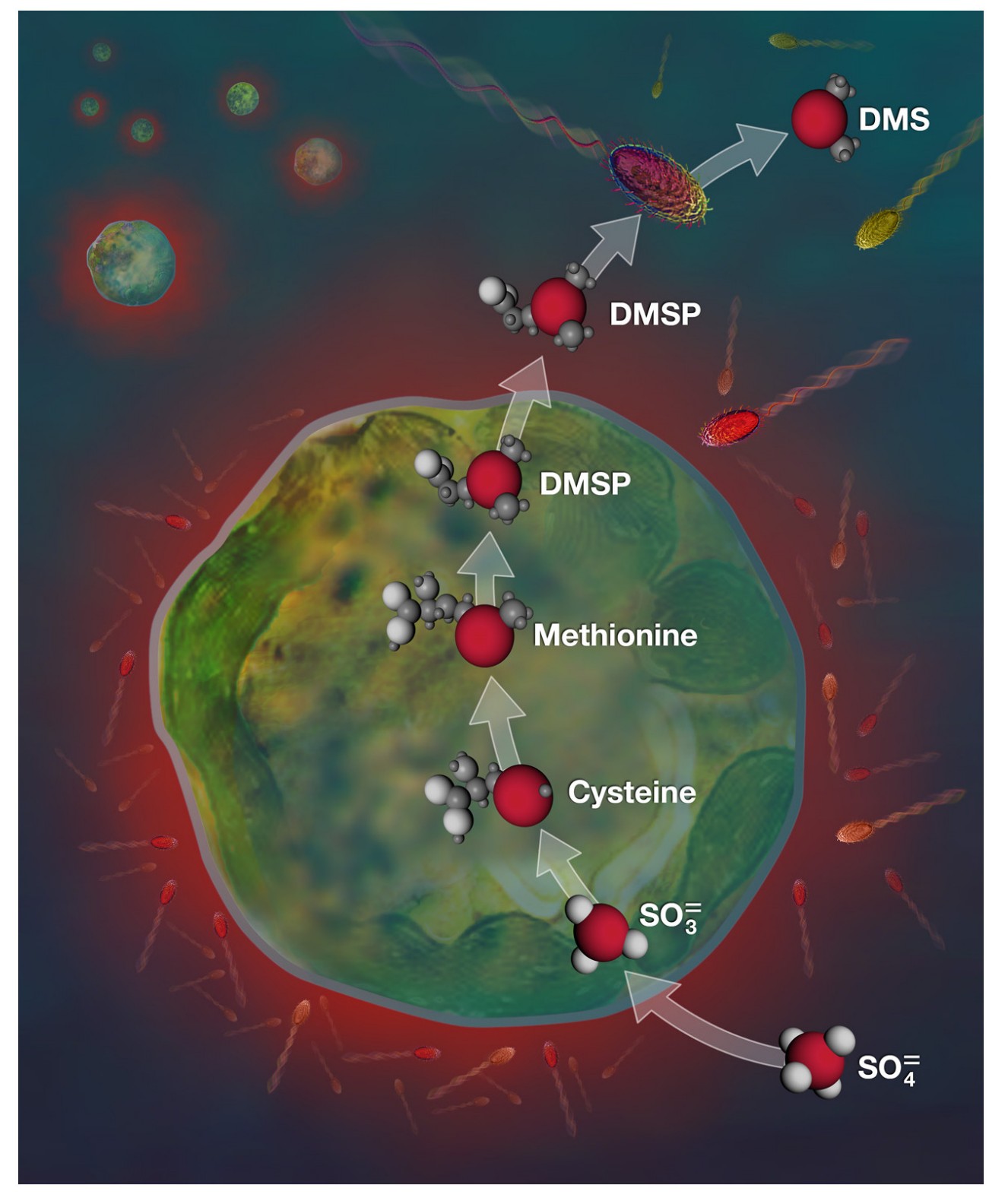

**Figure 1.** DMSP biosynthetic pathway targeted in this study. Sulfate ($SO_4^{2-}$) taken up from seawater by *Symbiodinium* is converted to sulfite ($SO_3^{2-}$), sulfur-based amino acids and finally DMSP. Some DMSP molecules are then exuded from *Symbiodinium* cells and can be degraded by a variety of marine bacteria (sulfur atoms (S) and bacterial cells that have taken up sulfur are in red). The biosynthetic pathway presented here is simplified, for more details see Stefels (*Stefels, 2000*).

*Figure 1 continued on next page*

*Figure 1 continued*

The following source data and figure supplements are available for figure 1:

**Source data 1.** ASP-8A supplement composition used for *Symbiodinium* cultures modified from *Blank (1987)*.
**Figure supplement 1.** Sampling design showing the four different culture treatments.
**Figure supplement 2.** Growth kinetics of *Symbiodinium* cells (strain C1; mean ± SE; *n* = 8) incubated at 27°C in artificial seawater containing either $^{34}SO_4^{2-}$ (red) or $^{nat}SO_4^{2-}$ (green) as the sole sulfur source.

sulfur for the diverse and highly abundant bacterial communities harboured by corals (*Raina et al., 2010*). Here we tracked and quantified the incorporation of a stable isotope of sulfur into *Symbiodinium* and its subsequent transfer to associated bacteria. To provide the first sub-cellular imaging and quantification of DMSP, we used a unique suite of analytical techniques, taking advantage of: (i) the spatial resolution afforded by nano-scale secondary ion mass spectrometry (NanoSIMS), (ii) the molecular characterization enabled by Time-of-Fight secondary ion mass spectrometry (ToF-SIMS), and (iii) the precise quantification allowed by nuclear magnetic resonance (NMR) and liquid chromatography-mass spectrometry (LC-MS).

## Results and discussion

We used the rare isotope $^{34}S$ as a tracer to follow the exchange of sulfur between marine microorganisms at the single-cell level. *Symbiodinium* cells were incubated for 18 days in artificial seawater containing $^{34}S$-labelled sulfate as the sole sulfur source ($^{34}S$-ASW; *Figure 1—figure supplement 1*). We relied exclusively on the *Symbiodinium* cellular machinery to biosynthesise and exude $^{34}S$-labelled DMSP following incubation with the $^{34}S$-sulfate precursor. To prevent direct uptake of $^{34}S$-sulfate by bacteria, all *Symbiodinium* cultures were rinsed thoroughly and re-inoculated into ASW containing sulfate in natural isotopic abundance ($^{nat}S$-ASW) before addition of bacterial cells. Two different bacterial strains were added to the rinsed cultures and co-incubated for six hours: (i) *Pseudovibrio* sp. P12, a DMSP-degrading bacterium isolated from healthy corals (*Raina et al., 2016*), selected because of its worldwide distribution in coastal waters (*Shieh et al., 2004*) and its abundance in benthic invertebrate communities (*Bondarev et al., 2013*); and (ii) a control, *Escherichia coli* W (ATCC 9637), a widely studied and fully sequenced strain, able to grow in seawater and not capable of degrading DMSP. To precisely localise bacterial cells, both strains were pre-grown in a medium enriched in the rare stable isotope $^{15}N$ (in amino-acids and ammonium form). The cellular incorporation of the stable isotope tracers ($^{34}S$ and $^{15}N$) was identified by an increase in the sulfur ($^{34}S/^{32}S$) and/or nitrogen ($^{15}N/^{14}N$) ratio above their natural abundance values (0.043 and 0.0037, respectively).

*Symbiodinium* cell numbers doubled during the incubation period in the medium containing $^{34}S$-labelled sulfate, reaching approximately 2.9 million cells ml$^{-1}$ after 18 days (*Figure 1—figure supplement 2*). LC-MS analyses carried out at the end of the experiment on extracted *Symbiodinium* cells confirmed that all cultures initially incubated with $^{34}S$-sulfate were highly enriched in $^{34}S$-DMSP, which represented up to 46% of the DMSP molecules present in samples analysed (*Figure 2*, *Figure 2—source data 1*). This result confirms that sulfur atoms used by dinoflagellates to synthesise DMSP can originate from the uptake of inorganic sulfate derived from seawater (*Stefels, 2000*). In addition to $^{34}S$-DMSP, unexpectedly high levels of $^{32}S$-DMSP (ranging from 54% to 66% of total DMSP) were recorded in *Symbiodinium* cultures (*Figure 2—source data 1*). The presence of these high levels of $^{32}S$-DMSP can be explained by a combination of two factors: (i) *Symbiodinium* cells density only doubled during the incubation phase in $^{34}S$-ASW, retaining a large fraction of the natural pool of $^{32}S$ initially present in the starting culture prior to the incubation; (ii) new $^{32}S$-DMSP might have been synthesised during the six hours immediately preceding sampling, when *Symbiodinium* cells were incubated in $^{nat}S$-ASW medium. Although high concentrations of DMSP were present in the methanolic *Symbiodinium* cells extract (*Figure 2—source data 1*), sulfur containing amino acids (methionine and cysteine) were not detected by LC-MS or $^1H$ NMR.

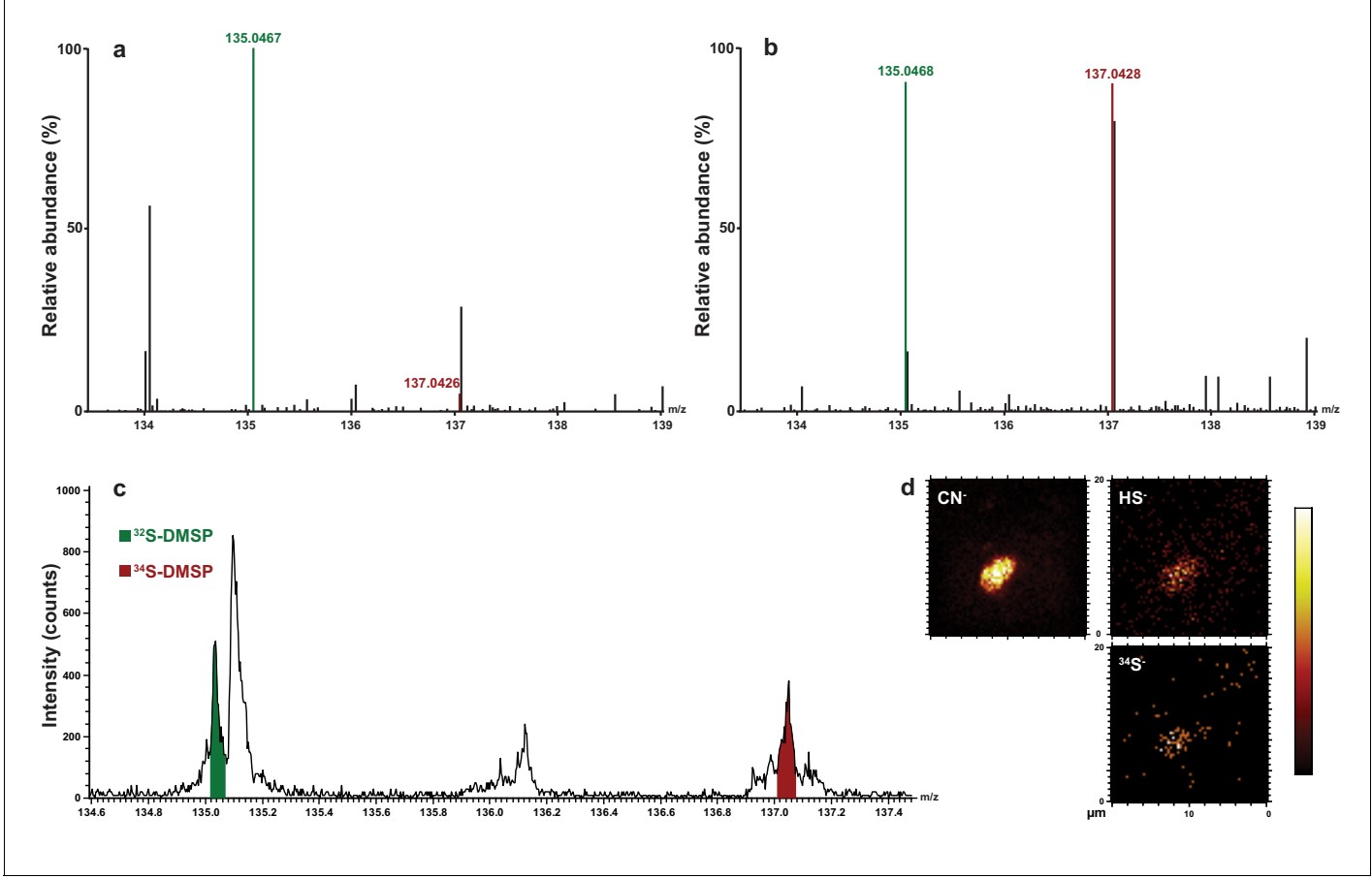

**Figure 2.** Representative HPLC-MS spectra showing the presence and relative abundance of $^{32}$S-DMSP (green peak) and $^{34}$S-DMSP (red peak) in methanol extracts derived from *Symbiodinium* culture (particulate fraction). (**a**) incubated with $^{nat}$S (treatment 4, see *Figure 1—figure supplement 1*); (**b**) incubated with $^{34}$S (treatment 3, see *Figure 1—figure supplement 1*). For more detailed spectra, see *Figure 2—figure supplement 2*; for absolute DMSP abundance, see *Figure 2—source data 1*. (**c**) Positive-ion ToF-SIMS spectrum of *Symbiodinium* incubated with $^{34}$S (treatment 3, see *Figure 1—figure supplement 1*) after resin embedding ($^{34}$S-DMSP represented 46% of total DMSP counts). For comparison between treatment and control spectra, see *Figure 2—figure supplement 1*; (**d**) Negative-ion ToF-SIMS images showing the distribution of CN⁻, HS⁻ and $^{34}$S⁻ species over a *Symbiodinium* cell (treatment 3, see *Figure 1—figure supplement 1*) enriched in $^{34}$S. Field of view is 20 × 20 µm² (lateral resolution is ~300 nm).

The following source data and figure supplements are available for figure 2:

**Source data 1.** DMSP in methanol extracts derived from the four different *Symbiodinium* culture treatments (particulate fraction), as measured by quantitative NMR (*n* = 3 biological replicates for cultures inoculated with *Pseudovibrio* sp.) and HPLC-MS ($^{32}$S-DMSP and $^{34}$S-DMSP fractions, *n* = 3).

**Figure supplement 1.** Representative positive-ion spectra of (a) Araldite 502 resin, and *Symbiodinium* (b) incubated with $^{nat}$S (treatment 4) and (c) incubated with $^{34}$S (treatment 3) after resin embedding.

**Figure supplement 2.** Representative HPLC-MS spectra showing the presence and relative abundance of $^{32}$S-DMSP (mass 135.04) and $^{34}$S-DMSP (mass 137.04) in methanol extracts: (a) DMSP standard containing natural abundance of $^{34}$S-DMSP; (b) *Symbiodinium* cells incubated with natS (treatment 4); (c) *Symbiodinium* cells incubated with $^{34}$S (treatment 3).

Up to 10% of the carbon fixed by photosynthetic algae is used for the production of DMSP (*Sievert et al., 2007*; *Archer et al., 2001*; *Simó et al., 2002*), which represents a major energy investment for these organisms and strongly suggests that this compound plays a central function in algal cells. To understand more precisely the functional role of DMSP, we used two SIMS approaches to infer its location within cells. To effectively prevent the loss of DMSP from the cells, the entire sampling procedure leading to SIMS analyses had to be water-free, with all steps performed under strict anhydrous conditions. For this, we used cryopreservation techniques followed by freeze

substitution in an acrolein-ether mixture. This method has routinely been used to successfully preserve cellular ions and compounds in a variety of systems (*Altus and Canny, 1985*; *Ashford et al., 1999*; *Kaiser et al., 2015*; *Marshall et al., 2007*; *Mostaert et al., 1996*), with the acrolein stabilizing and preserving cellular proteins, nucleic and fatty acids through cross linking, while the low temperature, anhydrous conditions ensure preservation and retention of diffusible ions and water-soluble molecules (such as DMSP). The inclusion of acrolein ensures excellent cell structural preservation at a low temperature, which is required for high resolution NanoSIMS analyses (*Kaiser et al., 2015*; *Marshall, 1980*).

ToF-SIMS revealed that $^{34}$S-DMSP was present and abundant in the preserved cells following resin embedding, with a ratio of $^{34}$S-DMSP/$^{32}$S-DMSP matching the bulk analyses carried out with LC-MS prior to embedding (*Figure 2c–d*, *Figure 2—figure supplement 2*). NanoSIMS analysis revealed that *Symbiodinium* exposed to $^{34}$S-labelled sulfate were nine times more enriched in $^{34}$S than the cells in the control ($^{34}$S/$^{32}$S ratio in $^{34}$S-ASW treatments: 0.391 ± 0.046, compared to $^{nat}$S-ASW controls 0.044 ± 0.001 [*Figure 4—figure supplement 1*]). Furthermore, substantial spatial variability in $^{34}$S enrichment was detected within *Symbiodinium* cells. Relatively low level of enrichments were detected in the nucleus ($^{34}$S/$^{32}$S: 0.087 ± 0.004) which might correspond to the presence of $^{34}$S-labelled amino-acids in the histone-like proteins that condense *Symbiodinium* DNA into chromosomes (*Shoguchi et al., 2013*) (*Figure 3*). Much higher enrichment levels were detected in vacuoles ($^{34}$S/$^{32}$S: 0.337 ± 0.011), chloroplasts ($^{34}$S/$^{32}$S: 0.384 ± 0.020) and cytoplasm ($^{34}$S/$^{32}$S: 0.451 ± 0.025); which means that the enrichment in these cellular structures was 7.7, 8.8 and 10.3 times over the natural abundance levels (*Figure 3*). However, the largest $^{34}$S enrichment was observed in small hotspots often observed near the *Symbiodinium* cell periphery ($^{34}$S/$^{32}$S: 0.971 ± 0.059; *Figure 3*), reaching more than 22 times the natural abundance level. Based on their small size and their high $^{34}$S enrichment, these hotpots are likely storage droplets containing sulfolipids, a group of sulfur compounds known to accumulate in *Symbiodinium* (*Garrett et al., 2013*; *Yuyama et al., 2016*). Lipid droplets of similar sizes and locations can be observed in these cells using electron microscopy (*Figure 3—figure supplement 1*). We were not able to detect methionine or cysteine using LC-MS or ToF-SIMS, which suggest that the intracellular concentration of these sulfur based amino-acids was relatively low. In contrast, DMSP is known to be by far the most abundant organic sulfur compound present in dinoflagellate cells (*Matrai and Keller, 1994*), representing more than 50% of the total organic sulfur in these organisms (*Matrai and Keller, 1994*). DMSP was the only organic sulfur compound we were able to detect in the *Symbiodinium* cells (through LC-MS, $^1$H NMR and ToF-SIMS), suggesting that most of the remaining $^{34}$S signal measured in *Symbiodinium* cells with NanoSIMS is highly likely originating from DMSP.

DMSP is an effective scavenger of reactive oxygen species (ROS), particularly hydroxyl radicals (•OH) (*Sunda et al., 2002*). The *in vivo* half-life of •OH is $10^{-9}$ seconds (*Sies, 1993*), which implies that these highly reactive molecules can damage lipids, nucleic acids, amino-acids or carbohydrates present in their direct vicinity. To be an effective antioxidant, a molecule needs not only to be able to scavenge ROS, but also to be located close to their source. Although the capacity of DMSP to detoxify ROS is established (*Sunda et al., 2002*), it has not been previously possible to ascertain its specific cellular function because its location is still unknown. If some DMSP is located in the cytoplasm, as suggested by our NanoSIMS data, it will be ideally localised to act as an osmolyte (*Kiene et al., 1996*). Furthermore, the presence of strong $^{34}$S signals in and around chloroplasts, where ROS are formed, support its role as an antioxidant (*Sunda et al., 2002*).

Following synthesis by phytoplankton, DMSP constitutes an important carbon and sulfur source for heterotrophic marine bacteria, which can either demethylate the compound and incorporate its sulfur into proteins or cleave it to produce DMS (*Curson et al., 2011*). At the termination of the experiment, total DMSP concentrations in *Symbiodinium* cells inoculated with the DMSP-degrading bacterium *Pseudovibrio* sp. P12 were 31% lower relative to those containing no bacteria or bacteria unable to degrade DMSP (*Figure 2—source data 1*). As *Symbiodinium* abundance did not differ between the treatments (*Figure 1—figure supplement 2*), the lower DMSP concentrations recorded are likely a consequence of the presence of *Pseudovibrio* cells able to degrade this compound. We sequenced the genome of *Pseudovibrio* sp. P12, revealing that this bacterium harbours a complete DMSP cleavage pathway, including a DMSP acyl-CoA transferase (encoded by *dddD*), a DMSP transporter (*dddT*) and the downstream catabolic enzymes (*dddB-C*) (*Todd et al., 2007*; *Raina et al., 2016*). Further analyses using NMR revealed that this DMSP degradation pathway was functional,

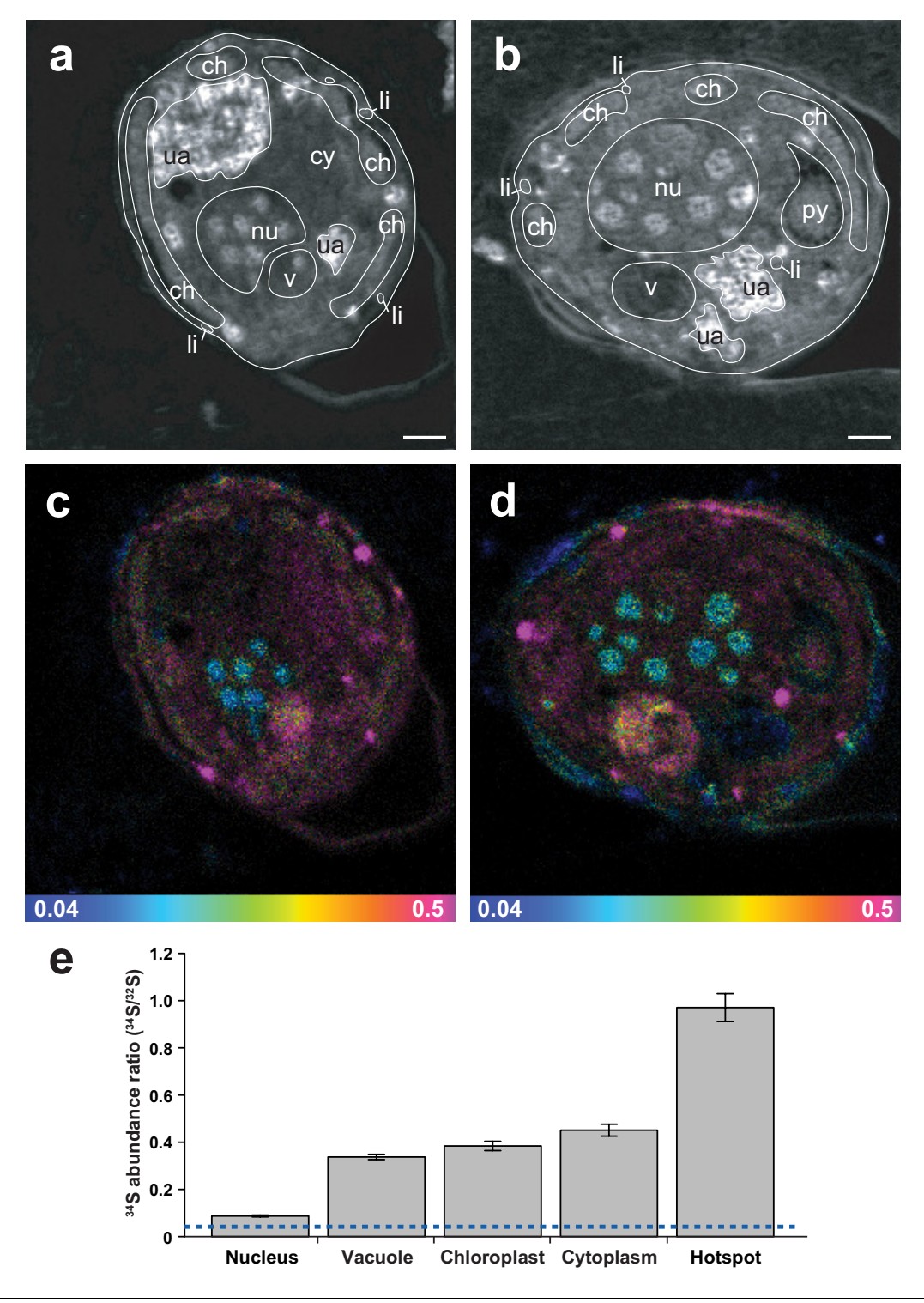

**Figure 3.** Representative NanoSIMS ion images of *Symbiodinium* cells showing the sub-cellular distribution of $^{34}$S. (a and b) $^{12}$C$^{14}$N/$^{12}$C$_2$ mass images showing cellular structures. (c and d) $^{34}$S/$^{32}$S ratio images of the same cells, shown as Hue Saturation Intensity (HSI) images where the colour scale indicates the value of the $^{34}$S/$^{32}$S ratio, with natural abundance in blue, changing to pink with increasing $^{34}$S levels. (e) Isotope ratio of $^{34}$S/$^{32}$S in different cellular regions (nucleus $n = 10$; vacuole $n = 3$; chloroplast $n = 35$; cytoplasm $n = 12$; hotspot $n = 20$; error bar: SE; source data available: *Figure 3—source data 1*). The dashed blue line represents the natural $^{34}$S abundance recorded in the control samples. nu: nucleus; ch: chloroplast; py: pyrenoid; ua: uric acid storage; v: vacuole; cy: cytoplasm; li: sulfolipids. Scale bars: 1 μm.

*Figure 3 continued on next page*

*Figure 3 continued*

The following source data and figure supplement are available for figure 3:

**Source data 1.** $^{32}$S and $^{34}$S measured in the different cellular region depicted in *Figure 3e*.
**Figure supplement 1.** Representative electron micrographs of *Symbiodinium* cells after OsO$_4$ staining showing the position and size of intracellular lipid droplets.

enabling this strain to convert high concentrations of DMSP into DMS (*Raina et al., 2016*). In addition, *Pseudovibrio* sp. P12 harbours homologues of genes involved in the demethylation pathway (*dmdA-B-C-D*), though these genes have a relatively low sequence identity (24%, 30%, 43% and 32%, respectively) (*Raina et al., 2016*) to the genes originally identified in *Ruegeria pomeroyi* DSS-3 (*Reisch et al., 2011*).

Bacteria-sized $^{15}$N hotspots localised outside *Symbiodinium* cells in NanoSIMS images were accurately identified as inoculated bacterial cells based on their unique nitrogen isotopic signatures (1151-fold increase on average over natural abundance, $n = 79$, *Figure 4—figure supplement 1*). Notably, within the *Pseudovibrio* treatment, the position of these $^{15}$N hotspots correlated exactly with $^{34}$S hotspots (*Figure 4*), which were characterised by a 3.3-fold increase in the $^{34}$S/$^{32}$S ratio over natural abundance ($n = 60$, *Figure 4h*). These observations confirmed that *Pseudovibrio* cells assimilated $^{34}$S-labelled *Symbiodinium*-derived metabolites. A 34% increase was also recorded in the mean $^{34}$S/$^{32}$S ratio of *E. coli* cells ($0.058 \pm 0.002$; $n = 19$), which are unable to degrade DMSP (compared to controls: 0.0438, *Figure 4h*). This enrichment, significantly higher than the expected natural abundance levels (*t*-Test, $n = 19$, $t = 9.227$, *p<0.001), can be explained by: (i) the capacity of *E. coli* to uptake small quantities of DMSP through betaine transporters to use as an osmoprotectant (*Cosquer et al., 1999*); (ii) the exudation of small quantities of other sulfur-containing substrates by *Symbiodinium*, such as methionine, which occur at a ratio of $8.2 \pm 2.6$ per 1000 amino acid residues in these dinoflagellates (*Markell and Trench, 1993*). In contrast, the high $^{34}$S enrichment recorded in *Pseudovibrio* cells, together with the significant decrease of particulate DMSP recorded in *Pseudovibrio*-inoculated treatments (*Figure 4i*), are likely due to the incorporation and degradation of DMSP. A comparison of $^{34}$S uptake between the two bacterial strains further highlights differences in their capacity to metabolise DMSP; *Pseudovibrio* incorporated seven times more sulfur than *E. coli* during the six-hours incubation (*Pseudovibrio*: specific uptake of $6.4 \pm 0.3$ ng S mg$^{-1}$ of dry weight, $n = 60$; *E. coli*: $0.9 \pm 0.1$ ng S mg$^{-1}$ of dry weight, $n = 19$). However, enzymatic cleavage of $^{34}$S-DMSP into volatile $^{34}$S-DMS, which diffuses out of *Pseudovibrio* cells and is therefore not captured by our NanoSIMS measurements, are likely to have caused an underestimation of the amount of sulfur cycled by this bacterium.

The marine sulfur cycle is a fundamental driver of atmospheric chemistry and climatic processes, yet its global influence is the product of unquantified cellular interactions between microorganisms. Here we used two SIMS approaches to directly visualise the accumulation and subsequent transfer of DMSP between marine microalgae and bacteria with unprecedented sub-cellular resolution. We applied a method that enables the preservation of water-soluble compounds, such as DMSP, in samples. This procedure, applicable to any system, may serve as a template to study the sub-cellular localization and identification of other small and highly diffusible molecules. In addition, similarly to other recent stable isotope approaches (*Stefels et al., 2009*), our method may be used to quantify the production rate of DMSP at the single cell level. We confirmed that $^{34}$S-DMSP was the main organic sulfur compound within the algal cells and we subsequently localised large quantities of the sulfur tracer $^{34}$S in algal vacuole, cytoplasm and chloroplasts. This strongly indicates that the relative concentrations of DMSP are higher in these key cellular locations, providing corroborative evidence for its functional role in mitigating both osmotic and oxidative stresses. Taken together, we have demonstrated that it is possible to image and quantify DMSP in phytoplankton and their associated bacteria at the sub-cellular scale. These methods open the way to further studies resolving the role of DMSP in phytoplankton and its contribution to phytoplankton-bacteria interactions.

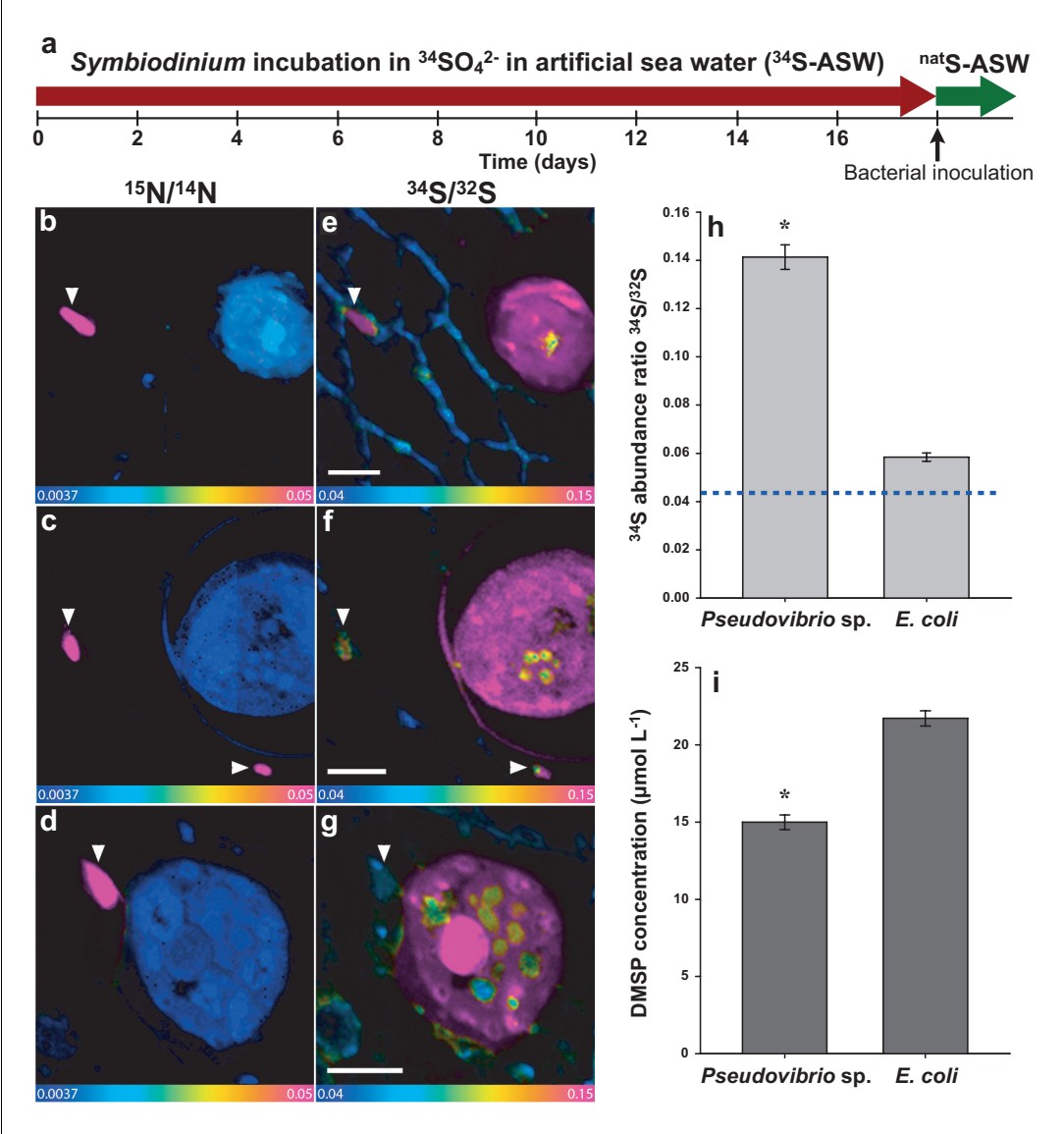

**Figure 4.** Representative NanoSIMS ion images of *Symbiodinium* cells exposed to $^{34}$S- or $^{nat}$S-artificial seawater (ASW) for 18 days and subsequently inoculated with two different bacterial strains for six hours. (a) Timeline of the experiment. (b, c and d) $^{12}C^{15}N/^{12}C^{14}N$ mass images showing the presence of $^{15}$N enriched bacterial cells. (e, f and g) $^{34}S/^{32}S$ ratio image of the same regions. These mass images are shown as HSI images where the colour scale indicates the value of the stable isotope ratios, with natural abundance in blue, changing to pink with increasing $^{15}$N or $^{34}$S levels. (b, c, e and f) *Symbiodinium* cultures were inoculated with the DMSP-degrading bacterium *Pseudovibrio* sp. P12 (treatment 1). (d and g) *Symbiodinium* cultures were inoculated with *Escherichia coli* (treatment 2). White arrows indicate bacteria. (h) Isotope ratio of $^{34}S/^{32}S$ in bacteria, *Pseudovibrio* cells were significantly more enriched than *E. coli* (t-Test, n = 60, t = 9.021, *p<0.001, error bars: SE). The dashed blue line represents the natural $^{34}$S abundance recorded in the control samples. (i) Total particulate DMSP concentration in *Symbiodinium* inoculated with *Pseudovibrio* sp. or *E. coli* (t-Test, n = 3, t = 9.908, *p<0.001, error bar: SE). Source data available: *Figure 4—source data 1*. Note: two regions of interest were merged to create *Figure 4c* due to stage-shifting errors during sequential acquisition of N and S data. Scale bars = 3 μm.

The following source data and figure supplement are available for figure 4:

**Source data 1.** $^{12}C^{15}N$, $^{12}C^{14}N$, $^{32}$S and $^{34}$S measured in the different organisms and treatments depicted in *Figure 4h* and *Figure 4—figure supplement 1*.

**Figure supplement 1.** Isotope ratio of (a) $^{15}N/^{14}N$ and (b) $^{34}S/^{32}S$ in *Symbiodinium* and bacteria cells measured by NanoSIMS in the different treatments (values were extracted from the images).

## Materials and methods

### Isolation of *Symbiodinium* and bacteria

Cells of *Symbiodinium* type C1 (confirmed by sequencing of the ITS1 gene) used in this study were isolated from air-brushed tissues of the coral *Acropora tenuis*, which had been collected from Magnetic Island, Great Barrier Reef, Australia (latitude 19°10'S; longitude 146°50'E). Cells were sequentially washed three times (5 min at 1600 g) with 0.2 μm filtered seawater. Clean *Symbiodinium* cells were inoculated into 24 well plates with sterile IMK medium (Wako Chemicals, Richmond, VA, USA) with the antibiotics penicillin (100 μg ml$^{-1}$), neomycin (100 μg ml$^{-1}$), streptomycin (100 μg ml$^{-1}$), nystatin (100 μg ml$^{-1}$), amphotericin (2.5 μg ml$^{-1}$), and Germanium dioxide (50 μM)) for 15 days at 27°C, 50 μE and 14:10 light:dark cycle. After this initial incubation, cells from uncontaminated wells were pooled and re-inoculated in new 24-well plates with IMK medium plus antibiotics as above, and incubated for 20 days at the same temperature and lighting conditions. Finally, uncontaminated cells were pooled and inoculated into 25 mL of sterile IMK without antibiotics until the start of the experiment (*Santos et al., 2011*). Cultures were genotyped by single-strand conformation polymorphism (SSCP) of the ITS1 region (*van Oppen et al., 2001*).

A coral-associated bacterium, *Pseudovibrio* sp. P12, was isolated from healthy colonies of the reef-building coral *Pocillopora damicornis*. This bacterial strain is commonly associated with reef-building corals (*Bondarev et al., 2013*; *Nissimov et al., 2009*; *Radjasa et al., 2008*; *Ritchie, 2006*; *Rypien et al., 2010*; *Sulistiyani et al., 2010*) and capable of metabolizing DMSP as a sole carbon source (*Garren et al., 2014*). Coral colonies were collected from Davies Reef, Great Barrier Reef, Australia (latitude 18°51'S; longitude 147°41'E) and maintained in aquaria at the Australian Institute of Marine Science (Townsville, Queensland, Australia) prior to strain isolation. A dilution series of coral tissue slurries was inoculated on minimal marine agar plates (1% bacteriological agar; 0.3% casamino acids; 0.4% glucose; in 1 litre of artificial seawater) (*Hjelm et al., 2004*). After 2 days of incubation at 28°C, single colonies were transferred into Marine Broth (Difco) and grown overnight. Liquid cultures were re-plated on minimal marine agar and the procedure was repeated iteratively until pure cultures were obtained. A laboratory strain of *Escherichia coli* (*E. coli* W (ATCC 9637)) was chosen as a control strain based on its ability to grow in the artificial seawater used in this study (see medium composition below), and its lack of DMSP degradation and subsequent sulfur assimilation pathways (unlike many marine or coral bacterial isolates [*Raina et al., 2010*; *Howard et al., 2008*]).

### Bacterial genomic analysis

High molecular weight DNA from a pure culture of the *Pseudovibrio* sp. P12 strain was obtained using a miniprep phenol/chloroform based DNA extraction (*Ausubel et al., 1987*). A paired-end library was prepared using the Illumina Truseq protocol (Illimina, San Diego, CA, USA), with an insert size of 169 bp and a read size of 150 bp. The library was sequenced on an Illumina MiSeq instrument at Monash University (Melbourne, Australia). The genome was assembled with the SPAdes assembler (v2.4.0) (*Bankevich et al., 2012*) and annotated with the Prokka software (v1.5.2) (*Seemann, 2014*), providing a draft genome assembly of *Pseudovibrio* sp. P12. The presence of the genes involved in DMSP metabolism was investigated by searching for homologs of the corresponding genes using reciprocal best BLAST hits.

### Synthesis of labelled magnesium sulfate (Mg$^{34}$SO$_4$)

Magnesium sulfate (Mg$^{34}$SO$_4$) was synthesised from pure sulfur $^{34}$S (purity >99%, Cambridge Isotope, MA) following a two-step reaction:

1. $6HNO_3 + {}^{34}S \rightarrow H_2{}^{34}SO_4 + 6NO_2 + 2H_2O$
2. $H_2{}^{34}SO_4 + MgCO_3 \rightarrow Mg{}^{34}SO_4 + H_2O + CO_2$

Elemental sulfur $^{34}$S (0.1069 g) was ground into a fine powder and transferred to a pear-shaped flask. Nitric acid (65%, 4 ml) was added to the flask, heated to 80°C and refluxed for 5 hr. The temperature was subsequently raised to 130°C and refluxed for an additional 24 hr in order to completely oxidise remaining nitric acid. The resulting sulfuric acid (H$_2$$^{34}$SO$_4$) was then converted to Mg$^{34}$SO$_4$ by the addition of magnesium carbonate (MgCO$_{3)}$ (0.2643 g), giving a yield of 0.3780 g. The solution was subsequently heated to 100°C until all water had completely evaporated. Elemental

analysis of the dried crystals was carried out with an electron probe microanalyser (EPMA, Jeol JXA8200), equipped with an energy dispersive spectrometer (EDS), to confirm the synthesis of $Mg^{34}SO_4$.

## *Symbiodinium* growth and experimental conditions

*Symbiodinium* C1 cells were inoculated into artificial seawater (ASW; starting density: $1.5 \times 10^6$ cells $ml^{-1}$) and incubated at 27°C for 18 days (based on results from a pilot study). LED lights were mounted above the culture, providing an average light intensity of 50 µE over a 14:10 hr light/dark cycle (AI Super Blue LED module 1003, IA, USA). Temperature and light intensities were monitored every 2 min for the duration of the experiment (using a HOBO UA-002-64, 64K temperature/light data logger).

The ASW contained 24.72 g of NaCl, 0.67 g of KCl, 1.36 g of $CaCl_2 \cdot 2H_2O$, 4.66 g of $MgCl_2 \cdot 6H_2O$, 0.18 g of $NaHCO_3$, and 3.8 ml of modified ASP-8A solution (*Figure 1—source data 1*) in 1 litre of MilliQ water. Magnesium sulfate ($MgSO_4 \cdot 7H_2O$, 6.29 g $L^{-1}$) was used as the sole sulfur source, with either $^{34}S$ (99% $^{34}S$, hereafter called $^{34}S$-ASW) or natural abundance of sulfur (95% $^{32}S$, 0.7% $^{33}S$, 4.2% $^{34}S$; hereafter called $^{nat}S$-ASW). *Symbiodinium* cells were incubated in $^{34}S$-ASW, whereas a batch incubated only in $^{nat}S$-ASW acted as a control. Both growth media were replaced every 5 days in order to actively remove dead and floating cells from the cultures. *Symbiodinium* cell numbers were monitored every 3 days for both $^{34}S$-ASW and $^{nat}S$-ASW treatments, using a light microscope and haemocytometer (depth 0.1 mm; eight replicates were averaged per time point) and cell mortality assessed using a 0.05% (w/v) Evans Blue solution (*Morera and Villanueva, 2009*).

After 18 days, the medium in both $^{34}S$-ASW and $^{nat}S$-ASW *Symbiodinium* cultures, was decanted and discarded. The *Symbiodinium* cells were thoroughly rinsed three times with $^{nat}S$-ASW and subsequently resuspended in $^{nat}S$-ASW (5 mins) prior to the addition of bacteria (*Figure 1—figure supplement 1*). This medium exchange (from $^{34}S$-ASW to $^{nat}S$-ASW) was carried out in order to prevent any potential direct bacterial uptake of $^{34}SO_4^{2-}$.

## Bacterial growth and inoculation

The two bacterial strains (*Pseudovibrio* sp. P12 and *E. coli* W) were grown overnight at 28°C in ASW medium enriched with $^{15}N$ (in the form of amino-acids and $NH_4^+$; Celtone Base Powder; Cambridge Isotope Laboratories, Tewksbury, MA). The bacterial cells were subsequently washed three times in ASW before inoculation. *Symbiodinium* cells in treatment 1 were subsequently inoculated with the DMSP-degrading bacterium *Pseudovibrio* sp. P12; treatment 2 with *E. coli*; treatment 3 acted as a control without bacteria added; and treatment 4, which had no contact with $^{34}S$, acted as negative control for sulfur isotope incorporation (*Figure 1—figure supplement 1*). The two bacterial strains were inoculated at a density of $10^6$ cells $ml^{-1}$ and all samples were collected six hours after bacterial inoculation (based on results from a pilot study).

## Sample preparation for NanoSIMS, electron microscopy and ToF-SIMS

We used high-pressure freezing (*Smart et al., 2010*), followed by a water-free embedding procedure to effectively prevent the loss of highly soluble compounds such as DMSP from our samples. This method does retain elements in solution (*Altus and Canny, 1985*; *Ashford et al., 1999*; *Kaiser et al., 2015*; *Marshall et al., 2007*; *Mostaert et al., 1996*) by effectively replacing the 'solution' with resin, without displacing the ions and osmolytes. *Symbiodinium* cultures pre-incubated with bacteria (20 µl) were dropped onto Thermanox strips (Thermo Fisher Scientific, Waltham, MA, USA, 4 × 18 mm) and then placed in humidified chambers. After 15 min, the cells settled onto the strips and the excess medium was carefully removed with filter paper before being frozen by immersion into liquid nitrogen slush (liquid nitrogen placed under low-vacuum in order to lower its temperature). Samples for structural imaging by electron microscopy (2 µl) were also collected. These were deposited in a gold planchet and high-pressure frozen using an EMPACT2 high-pressure freezer (Leica Microsystems, Wetzlar, Germany). Both sample types were stored in liquid nitrogen until required.

Frozen samples for NanoSIMS were freeze-substituted in anhydrous 10% acrolein in diethyl ether, and warmed progressively to room temperature over three weeks in an EM AFS2 automatic freeze-substitution unit (Leica Microsystems, Wetzlar, Germany) based upon the original method of

Marshall (*Marshall, 1980*), and as described recently in step-by-step detail by Kilburn and Clode (*Kilburn and Clode, 2014*). The samples were subsequently infiltrated and embedded in anhydrous Araldite 502 resin, after which the Thermanox strip was removed and the sample re-embedded and stored in a desiccator. Although it is possible that not 100% of cellular DMSP may be preserved by this procedure, any losses would affect all samples equally; not impacting the validity of our comparisons between treatments. Furthermore, as $^{15}N$ was only used as a tag to visualise the bacteria, dilution by processing and resin embedding (*Musat et al., 2014*) is of no concern here. For $^{34}S$ analyses, dilution can be expected to be negligible as there is no sulfur contained in processing or resin components. Resin sections (1 μm thick) of embedded *Symbiodinium* cells were cut dry using a Diatome-Histo diamond knife on an EM UC6 Ultramicrotome (Leica Microsystems, Wetzlar, Germany), mounted on a silicon wafer and coated with 5 nm of gold.

## NanoSIMS analysis

The NanoSIMS-50 (Cameca, Gennevilliers, France) at the Centre for Microscopy, Characterisation and Analysis (CMCA) at The University of Western Australia was used for all subsequent analyses. The NanoSIMS-50 allows simultaneous collection and counting of multiple isotopic species, which enables the determination of $^{15}N/^{14}N$ and $^{34}S/^{32}S$ ratios. Enrichments of the rare isotopes $^{34}S$ and $^{15}N$ were confirmed by an increase in the sulfur ($^{34}S/^{32}S$) and/or nitrogen ($^{15}N/^{14}N$) ratio above natural abundance values recorded in controls (equal to 0.0438 and 0.00367, respectively).

NanoSIMS analysis was undertaken by rastering a 2 pA $Cs^+$ beam (~100 nm diameter) across defined 20 μm$^2$ sample areas (256 × 256 pixels). The NanoSIMS-50 was tuned to achieve mass resolution at levels where the isobaric species $^{12}C^{15}N$ and $^{13}C^{14}N$ could be separated. The isotope ratio values are represented hereafter using a colour-coded transform (hue saturation intensity (HSI)) showing natural abundance levels in blue, and grading to high enrichment in pink. Images were processed and analysed using Fiji (http://fiji.sc/Fiji) (*Schindelin et al., 2012*) with the Open-MIMS plug-in (http://nrims.harvard.edu/software). All images were dead-time corrected (*Hillion et al., 2008*). Quantitative data were extracted from the mass images through manually drawn regions of interest. Ratio data were tested for QSA (quasi-simultaneous arrivals) by applying different beta values from 0.5 to 1[62]. No differences in the data were observed, indicating that the secondary ion count rates were too low to be affected by QSA.

## Time-of-flight secondary ion mass spectrometry (ToF-SIMS)

During ToF-SIMS analysis the sample surface is sputtered with a focused primary ion beam to produce ionic species (secondary ions) of the atoms, molecules and molecular fragments from the uppermost monolayers of the surface. The secondary ions are extracted into a flight column (time-of-flight analyser) and their masses determined by the exact time at which they arrive at the detector. The data collected can provide: (i) mass spectral information in the form of an accumulated mass spectrum, and (ii) image information in the XY dimensions showing the intensity distribution of the specific secondary ions from the area analysed.

The mass resolution of the ToF-SIMS analysis is determined by the temporal pulse width of the primary ions hitting the sample surface; whereas the spatial resolution is determined by the spot size of the primary ion beam. ToF-SIMS analyses are conducted with the instruments optimised either for high mass resolution or for high spatial resolution, as achieving both short pulses (for mass resolution) and narrow focus (for spatial resolution) simultaneously will greatly reduce the primary ion current density.

In this study, ToF-SIMS analyses were conducted using the TOF.SIMS five instrument (ION-TOF GmbH, Münster, Germany) at the Mark Wainwright Analytical Centre (MWAC), University of New South Wales. The instrument is equipped with a bismuth liquid metal cluster ion gun for analysis and an electron flood gun for charge compensation. Analysis was performed using a 30 keV $Bi_3^+$ cluster ion beam on resin sections (1 μm thick) mounted on silicon wafers. The 'spectrometry' mode was used to acquire high-mass resolution spectra ($m/\triangle m > 4000$) and 'fast-imaging' mode was used to acquire high spatial resolution images (lateral resolution ~300 nm, $m/\Delta m \sim 200$).

In a typical analysis, a positive ion spectrum was acquired over a defined area of 20 × 20 μm$^2$ or 50 × 50 μm$^2$. The area of interest was identified by negative ion images acquired over areas of 20 × 20 μm$^2$ (64 × 64 pixels) to 200 × 200 μm$^2$ (128 × 128 pixels), where maps of CN$^-$ ($m/z$ 26), S$^-$

($m/z$ 32), HS$^-$ ($m/z$ 33) and $^{34}$S$^-$ ($m/z$ 34) were generated to locate the position of cells and the presence of sulfur-containing compounds within the sample. Care was taken to ensure the ion dose density was kept below the static SIMS limit ($10^{12}$–$10^{13}$ primary ions per cm$^2$) (*Lindgren et al., 2014*) when acquiring imaging data, *e.g.* no more than 5–10 scans over areas of 20 × 20 μm$^2$. Keeping the static limit in the imaging mode prevents any significant damage to the sample structure or chemistry (*Vickerman and Briggs, 2013*), and enables further analyses of the same area in the positive polarity in this case. In the positive spectrum, the molecular ion [M + H]$^+$ peak of both the $^{32}$S-and $^{34}$S-containing DMSP molecules (C$_5$H$_{11}$$^{32}$SO$_2$$^+$ and C$_5$H$_{11}$$^{34}$SO$_2$$^+$, respectively) are closely spaced with peaks arise from the resin (*Figure 2—figure supplement 2*). To maximise signal-to-noise ratio, data acquisition over a relatively small area encompassing the cell was desired, allowing unambiguous identification of the C$_5$H$_{11}$$^{32}$SO$_2$$^+$ and C$_5$H$_{11}$$^{34}$SO$_2$$^+$ peaks. High mass resolution positive spectra were calibrated using the masses of CH$_2$$^+$, C$_2$H$_4$$^+$, C$_4$H$_8$$^+$ and C$_6$H$_{12}$$^+$ molecules. Data processing and evaluation were conducted using the SurfaceLab six software package (ION-TOF GmbH, Münster, Germany).

Prior to the analyses of the resin sections, the mass spectrum of dimethyl-$\beta$-propiothetin standard (Research Plus Inc., USA) was recorded to provide spectral information of DMSP generated by ToF-SIMS analysis. The molecular ion [M + H]+ peak (C5H11SO2+, m/z 135.05) was observed to be the most intense peak in the spectrum, and was used as the mass peak position when determining the presence of DMSP molecules in the samples. The mass spectrum of a mixture of methionine and cysteine (Sigma-Aldrich, USA) was also acquired to serve as a reference standard. Both methionine and cysteine were not detected or the amounts were below the detection limit of the instrument (ppm range).

## Transmission electron microscopy (TEM)

High-pressure frozen samples for structural imaging were freeze-substituted in 1% OsO$_4$ in acetone over two days and similarly infiltrated and embedded as described above. Sections 90 nm thick were cut on water using a diamond knife, collected on copper grids and imaged unstained at 120 kV in a JEOL 2100 TEM (Tokyo, Japan) fitted with a Gatan ORIUS camera (California, USA). Please note: the high solubility of DMSP in water prevented the coupling of NanoSIMS with TEM images (*Clode et al., 2009*) to identify the location of small organelles such as mitochondria, as ultrathin sections cannot be prepared without exposing the samples to water.

## High pressure liquid chromatography-mass spectrometry (HPLC-MS)

After samples were collected for NanoSIMS analysis, all *Symbiodinium* cultures were centrifuged (3000 g) for 5 min, the medium was discarded and pelleted cells were extracted with 5 mL of HPLC-grade methanol. Crude methanol extracts were then analysed by reverse-phase (RP18) HPLC-MS in triplicate along with pure DMSP and amino acid standards.

A 10 μL aliquot of the methanol extract was diluted with an equal volume of acetonitrile and chromatographed using a Waters Alliance 2695 HPLC system comprising a quaternary pump, autosampler and photodiode array detector (200–400 nm) coupled to a Waters Micromass LCT Premier orthogonal acceleration time-of-flight (oa-TOF) mass spectrometer. Separation was achieved on an Alltima HP HILIC column (250 × 4.6 mm with a particle size of 5 μm) at 27°C and a flow rate of 0.75 ml min-1. The gradient was: acetonitrile (90%):0.1% formic acid (10%) at 0 min; acetonitrile (60%):0.1% formic acid (40%) at 0.4 min; acetonitrile (10%):0.1% formic acid (90%) at 12 min; acetonitrile (90%):0.1% formic acid (10%) at 12.25 min.

TOF-MS accurate mass measurements (scan-range $m/z$ 100–1000 at 4 GHz, resolution = 9500) were acquired using an electrospray ionization (ESI) source in W positive mode with the following operation parameters: capillary voltage: 3000 V; cone voltage: 80V; ion source temperature: 80°C; desolvation temperature: 350°C; cone gas flow: 10 l hr$^{-1}$; desolvation gas flow: 750 l hr$^{-1}$; ion energy: 33 V; acceleration voltage: 100 V. MassLynx software (version 4.1, Waters) was used for operating the HPLC-MS, as well as for data acquisition and processing. Leucine Enkephalin was used as the external reference.

## Quantitative nuclear magnetic resonance (qNMR)

The MeOH extracts remaining after HPLC-MS analysis was dried using a vacuum-centrifuge and dissolved in a mixture of deuterium oxide ($D_2O$, D 99.8%, 250 µl) and deuterated methanol ($CD_3OD$, D 99.8%, 750 µl) (Cambridge Isotope Laboratories, Andover, MA, USA). A 700 µl aliquot of the particulate-free extract was transferred into a 5 mm Norell 509-UP-7 NMR tube (Norell Inc., Landisville, NJ, USA) and analysed immediately by [1]H NMR.

[1]H NMR spectra were recorded on a Bruker Avance 600 MHz NMR spectrometer with TXI 5 mm probe and quantification performed using the ERETIC method (*Tapiolas et al., 2013*). This technique generates an internal electronic reference signal, calibrated using stock solutions of DMSP.

## Sulfur uptake

Bacterial strains and *Symbiodinium* were counted (Becton Dickinson LSR II flow cytometer (BD Biosciences, Franklin Lakes, NJ, USA), and pellets were subsequently freeze-dried and weighed in order to determine their total sulfur content (equal to 5390 ng S $mg^{-1}$). Samples were analysed on a Thermo Scientific FLASH 2000 Series (Thermo Scientific, Waltham, MA, USA). The sulfur uptake per mg of bacterial cells ($\rho$) was expressed in ng S $mg^{-1}$ and was calculated by normalizing the [34]S-incorporation measured using NanoSIMS to the average sulfur content (% of dry mass) according to the equation of *Dugdale and Wilkerson (1986)*, presented in *Pernice et al. (2012)*.:

$\rho = ((S_{mes} - S_{nat})/(S_{enr} - S_{nat})) \times S_{content} \times 10^3$

Where:

$S_{mes}$: [34]S/[32]S measured in labelled samples by NanoSIMS

$S_{nat}$: natural abundance of [34]S/[32]S measured in unlabelled samples by NanoSIMS

$S_{enr}$: [34]S-enrichment of the *Symbiodinium* cells measured by NanoSIMS

$S_{content}$: average sulfur content (%) measured by Thermo Scientific FLASH 2000 Series.

The calculated uptake (in nmol S $mg^{-1}$) was then converted into an estimate uptake rate per day (nmol S $l^{-1}$ $day^{-1}$), based on: the bacterial exposure to [34]S (6 hr), and the bacterial cell density for a given dry weight (acquired through flow cytometry; equal to $7.12 \times 10^{-7}$ g for $5 \times 10^5$ bacterial cells).

# Acknowledgements

The authors would like to thank Sabina Belli for valuable comments on the manuscript. We specially thank John Cliff, Mike House, Sigrid Fraser, Jeremy Shaw, John Murphy, Lyn Kirilak, Brett Baillie, and Sara Bell for their technical assistance. We also thank Janja Ceh and Bernard O'Reilly for their indefectible support. Credit for *Figure 1*: Glynn Gorick, Jean-Baptiste Raina and Justin Seymour. This work was supported by ANNiMS (Australian Government, Department of Education, Employment and Workplace Relations), the AMMRF Centre for Microscopy, Characterisation and Analysis (UWA) and by Australian Research Council Grant DE160100636 to JBR.

# Additional information

## Funding

| Funder | Grant reference number | Author |
| --- | --- | --- |
| Australian Research Council | DE160100636 | Jean-Baptiste Raina |
| ANNiMS | | Jean-Baptiste Raina |
| AMMRF Centre for Microscopy, Characterisation and Analysis | | Jean-Baptiste Raina |

The funders had no role in study design, data collection and interpretation, or the decision to submit the work for publication.

## Author contributions

J-BR, Conceptualization, Data curation, Formal analysis, Funding acquisition, Methodology, Writing—original draft, Writing—review and editing; PLC, Conceptualization, Formal analysis,

Supervision, Validation, Methodology, Writing—original draft, Writing—review and editing; SC, For-mal analysis, Writing—review and editing; JB, AR, Formal analysis; MRK, Conceptualization, Formal analysis, Validation; SF, Resources, Formal analysis; MS, Resources, Investigation; VB, CEM, Resour-ces, Methodology; PT-H, Investigation, Methodology; DT, Conceptualization, Investigation; CMM, Conceptualization, Resources, Methodology; BG, Methodology; MP, Formal analysis, Investigation, Methodology; JRS, Resources, Supervision, Writing—original draft; BLW, Conceptualization, Super-vision, Writing—original draft; DGB, Conceptualization, Supervision, Writing—original draft, Writ-ing—review and editing

## Author ORCIDs

Jean-Baptiste Raina, http://orcid.org/0000-0002-7508-0004
Sylvain Forêt, http://orcid.org/0000-0002-4145-9243

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
