## [Decision Letter]

Thank you for submitting your article "Subcellular tracking reveals rapid dimethylsulfoniopropionate cycling among marine microbes" for consideration by *eLife*. Your article has been reviewed by two peer reviewers, and the evaluation has been overseen by a Reviewing Editor and Christian Hardtke as the Senior Editor. The following individual involved in review of your submission has agreed to reveal his identity: David Green (Reviewer #3).

The reviewers have discussed the reviews with one another and the Reviewing Editor has drafted this decision to help you prepare a revised submission.

Two reviewers have very carefully evaluated this paper and both came to similar conclusions that while the experiments are elegant, the results are over interpreted as rates. Clearly this issue is critical, as fluxes are not measured directly by a SIMS system, unless the authors used a pulse-chase approach (or a variant, such as reverse isotope dilution). Regardless, after reading the paper and the constructive comments offered by the reviewers, it is clear that, following a revision, the paper would potentially be suitable for publication in *eLife*. To that end, the authors should address the concerns of the reviewers, especially with respect to the issue of rates of DMSP cycling.

*Reviewer #2:*

This manuscript presents data on innovative experiments looking at the incorporation of ^[34]^S-sulfate into the cellular biomass of the dinoflagellate coral symbiont, *Symbiodinium*. The authors main focus is on the biosynthesis of dimethylsulfonioproprionate (DMSP), a major organic sulfur compound produced in large amounts by many marine phytoplankton, especially dinoflagellates. The authors used cutting-edge methods such as NanoSIMS, ToF-SIMS, and LCMS to identify DMSP and to localize sulfur in the cells. They also carried out an experiment in which bacteria were added to ^[34]^S-labeled cultures to look at the assimilation of ^[34]^S derived from the *Symbiodinium* by the bacteria. Both DMSP degrading and non-degrading bacteria were used, and the bacteria were pre-labeled with 15N to allow them to be confidently localized with the NanoSIMS.

The study represents amazing application of technology. I do however feel that some of the interpretations about DMSP localization need to be tempered. Based on the information provided, I don't think they can rule out the significant contribution of other organic sulfur compounds in the signals they observed with the NanoSIMS. The ToF-SIMS provided some identification of ^[34]^S-DMSP in the cells but it wasn't clear to me whether they could exclude the occurrence of sulfur compounds, like cysteine and methionine, or a host of other possible organic sulfur compounds.

The authors assume that most of the cellular sulfur is DMSP. They cite an older study by Matrai and Keller that concluded that 50-90% of the cellular sulfur is DMSP. 50% might be reasonable, but 90% is probably not, given that there are many other organic sulfur compounds in living cells. I think the authors need to be more cautious in interpreting their data. This goes to my general criticism – that the authors are assuming most of the ^[34]^S associated with the cells is DMSP. It certainly seems that some of it is, and that is great. But they really haven't ruled out other sulfur compounds and the nanoSIMS doesn't distinguish what compounds the ^[34]^S is in. To support their argument that sulfur amino acids are not important, the authors say that "for *Symbiodinium*, methionine and cysteine occur at a ratio of 1 per 1000 amino acid residues" (Results and Discussion). They cite Markell and Trench (1993) for this information. This struck me as an extremely low sulfur amino acid content and it seems very unlikely given that sulfur amino acids are present in virtually all proteins, with cysteine being important in the active site of many enzymes, including DMSP lyase, which *Symbiodinium* produces. I looked at the Markell and Trench paper and I found that the authors have misrepresented the earlier work in the way they cited it. First, the data in Markell and Trench are for proteins in cellular exudates not necessarily the cells themselves; the present study is looking at the cells. Second, Markell and Trench report only Methionine data; cysteine was not reported, most likely because cysteine is difficult to quantify because of oxidation of the sulfhydryl group. Third, Markell and Trench looked at 5 different *Symbiodinium* strains and their Table 3 shows values for methionine per 1000 residues of 7, 7, 9, 17 and 1 in the 5 strains, respectively. So only one of the 5 strains had 1 methionine per 1000, with the others having about an order of magnitude more. And again, cysteine was not counted. So, there must be more sulfur in those cells besides DMSP. The data must be interpreted with this in mind. Overall, I am not convinced from the data presented that the authors have "confirmed that ^[34]^S-DMSP was the main organic sulfur compound within the algal cells" (Results and Discussion).

The title of the paper mentions rapid cycling of DMSP between microbes, but the results in this paper are not presented as rates. And I raise the point that some DMSP might have been released from cells as a result of the manipulations, so the rate at which transfer from the phytoplanker to the bacteria happens is still an open question.

It's not clear what the TOF-SIMS actually measures. I looked this up and I think I understand that it can give masses of fragments released by the ion beam and thus can provide some identification information. But I don't know enough about the specificity/selectivity/sensitivity of such an approach. They say that they observed ^[34]^S-DMSP this way. Were other compounds looked for? More details are needed.

The full reference for the Markell and Trench (1993) paper I cited in my review is:

Markell, D. A. & Trench, R. K. Macromolecules exuded by symbiotic dinoflagellates in culture: amino acid and sugar composition. J. Phycol. 29, 64-68 (1993).

I specifically referred to their Table 3 for Methionine data.

*Reviewer #3:*

This is an exciting and technically ground breaking report that helps nanoSIMS deliver the scientific potential it purportedly has. Understanding why DMSP is made by phytoplankton is one of the major challenges left in the DMSP story and this paper is very timely in demonstrating that it can be tracked at an intracellular level between phytoplankton and bacteria.

I am less sure this work has revealed any great understanding about how DMSP is made and consumed by bacteria, and certainly not its rate of production or how it is cycled. It is exciting as a milestone paper that proves the technique can be used to demonstrate important aspects of DMSP dynamics and for the data showing subcellular location. It will open the door to further hypothesis driven work by the authors and others to interrogate the synthesis and molecular roles of this molecule at the sub-cellular level.

I'm not sure the title reflects the real impact of this work. Stating it "…reveals rapid…" is misleading as the incubation period for DMSP uptake by bacteria was 6 hr – is that really rapid; and is this really what is important about this work? I feel that the sub-cellular location of DMSP in phytoplankton and bacteria is the important aspect and the title should reflect this more closely.

The authors have used the word "rate" and "cycling" in a number of places. I do not believe there is any data that reports rates of DMSP synthesis. I see rate methods (subsection “Sulfur uptake”, last paragraph etc.) and data bacterial uptake rates (e.g. Results and Discussion, seventh paragraph) – so I feel you need to be more specific and specify bacterial rates. And the cycling observed was the presence of DMSP in *Symbiodinium* (and sub-cell locations) and presence of DMSP in bacteria. Is this really cycling, or just a transfer and/or degradation step within a cycle?

I found that there was lots of good discussion and speculation about what the data might mean, but actually relatively little presentation and interpretation of the data (e.g. Results and Discussion, fifth paragraph). The authors need to redress this balance. In particular, data relating to the enrichment of S in the organelles is crying out for more interpretation – as this is new and you have brilliant nanoSIMS images of subcellular localisation that I do not feel have been mined anywhere near enough. If this is not possible, it is important to make this limitation clear in the text, as this would be an important limitation of the methodology or your use of it.

A comment: there was no mention of Stefels 2009 (Limnol. Oceanogr.: Methods 7, 2009, 595-611) work reporting use of stable isotopes to measure DMSP synthesis rates. Could the authors evaluate their approach to that of Stefels D2O approach – maybe comment whether nanoSIMS can be used in a similar way – as these are both important advances in methodology; somewhere around the last paragraph of the Results and Discussion?

It was difficult to follow which images were being referred to, as the individual image files were not labelled. This might be an oddity of the *eLife* submission system, I accept.

---

## [Author Response]

*Two reviewers have very carefully evaluated this paper and both came to similar conclusions that while the experiments are elegant, the results are over interpreted as rates. Clearly this issue is critical, as fluxes are not measured directly by a SIMS system, unless the authors used a pulse-chase approach (or a variant, such as reverse isotope dilution). Regardless, after reading the paper and the constructive comments offered by the reviewers, it is clear that, following a revision, the paper would potentially be suitable for publication in eLife. To that end, the authors should address the concerns of the reviewers, especially with respect to the issue of rates of DMSP cycling.*

The manuscript has been fully revised and all occurrences of the word rate have been removed from our data interpretation. We have clarified the methodology, discussed our data in detail (as requested by reviewer 3). We strongly believe that the revised manuscript addresses all the reviewer concerns.

*Reviewer #2:*

*[…] The study represents amazing application of technology. I do however feel that some of the interpretations about DMSP localization need to be tempered. Based on the information provided, I don't think they can rule out the significant contribution of other organic sulfur compounds in the signals they observed with the NanoSIMS. The ToF-SIMS provided some identification of ^[34]^S-DMSP in the cells but it wasn't clear to me whether they could exclude the occurrence of sulfur compounds, like cysteine and methionine, or a host of other possible organic sulfur compounds.*

*The authors assume that most of the cellular sulfur is DMSP. They cite an older study by Matrai and Keller that concluded that 50-90% of the cellular sulfur is DMSP. 50% might be reasonable, but 90% is probably not, given that there are many other organic sulfur compounds in living cells. I think the authors need to be more cautious in interpreting their data. This goes to my general criticism – that the authors are assuming most of the ^[34]^S associated with the cells is DMSP. It certainly seems that some of it is, and that is great. But they really haven't ruled out other sulfur compounds and the nanoSIMS doesn't distinguish what compounds the ^[34]^S is in. To support their argument that sulfur amino acids are not important, the authors say that "for Symbiodinium, methionine and cysteine occur at a ratio of 1 per 1000 amino acid residues" (Results and Discussion). They cite Markell and Trench (1993) for this information. This struck me as an extremely low sulfur amino acid content and it seems very unlikely given that sulfur amino acids are present in virtually all proteins, with cysteine being important in the active site of many enzymes, including DMSP lyase, which Symbiodinium produces. I looked at the Markell and Trench paper and I found that the authors have misrepresented the earlier work in the way they cited it. First, the data in Markell and Trench are for proteins in cellular exudates not necessarily the cells themselves; the present study is looking at the cells. Second, Markell and Trench report only Methionine data; cysteine was not reported, most likely because cysteine is difficult to quantify because of oxidation of the sulfhydryl group. Third, Markell and Trench looked at 5 different Symbiodinium strains and their* Table 3 *shows values for methionine per 1000 residues of 7, 7, 9, 17 and 1 in the 5 strains, respectively. So only one of the 5 strains had 1 methionine per 1000, with the others having about an order of magnitude more. And again, cysteine was not counted. So, there must be more sulfur in those cells besides DMSP. The data must be interpreted with this in mind. Overall, I am not convinced from the data presented that the authors have "confirmed that ^[34]^S-DMSP was the main organic sulfur compound within the algal cells" (Results and Discussion).*

The reviewer question our claim regarding the nature of the ^[34]^S signal detected in the cells via NanoSIMS, stating that: (1) “the authors are assuming most of the ^[34]^S associated with the cells is DMSP”; and (2) “to support their argument that sulfur amino acids are not important, the authors say that "for *Symbiodinium*, methionine and cysteine occur at a ratio of 1 per 1000 amino acid residues"”. However, these two points do not accurately depict what we have done:

1) We did not assume that most of the ^[34]^S is in DMSP form, we investigated the presence of ^[34]^S-DMSP, ^[34]^S-Cysteine and ^[34]^S-Methionine using ToF-SIMS in the embedded *Symbiodinium* cells and could only detect ^[34]^S-DMSP. This does not rule out the potential of other sulfur containing molecules being present, such as sulfo-lipids (see Results and Discussion, fifth paragraph). However, the data in this paper provides strong evidence that the levels of sulfur-containing amino-acids in these cells were much lower than DMSP. We have added three new paragraphs in the ToF-SIMS Methods section to clarify our methodology (subsection “Time-of-flight secondary ion mass spectrometry (ToF-SIMS)”).

2) We did not use Markell and Trench (1993) to support the claim that most ^[34]^S recorded in *Symbiodinium* is in DMSP form. We used this reference to explain the slight enrichment in ^[34]^S recorded in *E. coli*, a bacterium that cannot degrade DMSP. Therefore, we were not talking about cell contents here but specifically exuded sulfur compounds (which is precisely what Markell and Trench measured). Below is the text taken directly from our original submission:

“This enrichment can be explained by: (i) the capacity of *E. coli* to uptake small quantities of DMSP through betaine transporters to use as an osmoprotectant (Cosquer et al., 1999);

(ii) the exudation of small quantities of other sulfur-containing substrates by Symbiodinium, including methionine and cysteine, which occur at a ratio of 1 per 1000 amino acid residues in these dinoflagellates (Markell and Trench, 1993)”.

However, reviewer 2 had some highly valid concerns regarding the Table 3 presented in Markell and Trench (1993). There were indeed five different strains of *Symbiodinium* tested and the amount of cysteine in these species was not reported. We therefore modified the sentence as followed: “(ii) the exudation of small quantities of other sulfur-containing substrates by *Symbiodinium*, such as methionine, which occur at a ratio of 8.2 ± 2.6 per 1000 amino acid residues in these dinoflagellates”.

*The title of the paper mentions rapid cycling of DMSP between microbes, but the results in this paper are not presented as rates. And I raise the point that some DMSP might have been released from cells as a result of the manipulations, so the rate at which transfer from the phytoplanker to the bacteria happens is still an open question.*

We acknowledge this point and have changed the title of the manuscript accordingly. It now reads “Subcellular tracking reveals the location of dimethylsulfoniopropionate in microalgae and visualises its uptake by marine bacteria”. In addition, we have removed all the occurrence of the word “rate”.

*It's not clear what the TOF-SIMS actually measures. I looked this up and I think I understand that it can give masses of fragments released by the ion beam and thus can provide some identification information. But I don't know enough about the specificity/selectivity/sensitivity of such an approach. They say that they observed ^[34]^S-DMSP this way. Were other compounds looked for? More details are needed.*

*The full reference for the Markell and Trench (1993) paper I cited in my review is:*

*Markell, D. A. & Trench, R. K. Macromolecules exuded by symbiotic dinoflagellates in culture: amino acid and sugar composition. J. Phycol. 29, 64-68 (1993).*

*I specifically referred to their* Table 3 *for Methionine data.*

Three new paragraphs have been added to the ToF-SIMS Methods section. We have detailed how the technique works, its sensitivity as well as the other compounds that were looked for. We hope that this clarify the use of the technique, and provide all the requested information.

*Reviewer #3:*

*[…] I am less sure this work has revealed any great understanding about how DMSP is made and consumed by bacteria, and certainly not its rate of production or how it is cycled. It is exciting as a milestone paper that proves the technique can be used to demonstrate important aspects of DMSP dynamics and for the data showing subcellular location. It will open the door to further hypothesis driven work by the authors and others to interrogate the synthesis and molecular roles of this molecule at the sub-cellular level.*

*I'm not sure the title reflects the real impact of this work. Stating it "…reveals rapid…" is misleading as the incubation period for DMSP uptake by bacteria was 6 hr – is that really rapid; and is this really what is important about this work? I feel that the sub-cellular location of DMSP in phytoplankton and bacteria is the important aspect and the title should reflect this more closely.*

Our title has been changed to: “Subcellular tracking reveals the location of dimethylsulfoniopropionate in microalgae and visualises its uptake by marine bacteria”. We hope that the new title provides a better reflection of the manuscript.

*The authors have used the word "rate" and "cycling" in a number of places. I do not believe there is any data that reports rates of DMSP synthesis. I see rate methods (subsection “Sulfur uptake”, last paragraph etc.) and data bacterial uptake rates (e.g. Results and Discussion, seventh paragraph) – so I feel you need to be more specific and specify bacterial rates. And the cycling observed was the presence of DMSP in Symbiodinium (and sub-cell locations) and presence of DMSP in bacteria. Is this really cycling, or just a transfer and/or degradation step within a cycle?*

We agree with the reviewer and have removed all occurrences of the word “rate”.

*I found that there was lots of good discussion and speculation about what the data might mean, but actually relatively little presentation and interpretation of the data (e.g. Results and Discussion, fifth paragraph). The authors need to redress this balance. In particular, data relating to the enrichment of S in the organelles is crying out for more interpretation – as this is new and you have brilliant nanoSIMS images of subcellular localisation that I do not feel have been mined anywhere near enough. If this is not possible, it is important to make this limitation clear in the text, as this would be an important limitation of the methodology or your use of it.*

We have now added a new panel in Figure 3, breaking down the enrichment levels between all the different cellular compartments that can be safely identified in *Symbiodinium* cells (Figure 3). Following the reviewer recommendation, we have also added more interpretation of the data:

“Relatively low level of enrichments were detected in the nucleus (^[34]^S/^[22]^S: 0.087 ± 0.004) which might correspond to the presence of ^[34]^S-labelled amino-acids in the histone-like proteins that condensate *Symbiodinium* DNA into chromosomes (Garrett et al., 2003) (Figure 3). […] We were not able to detect methionine or cysteine using LC-MS or ToF-SIMS, which

suggest that the intracellular concentration of these sulfur based amino-acids was relatively low.”

*A comment: there was no mention of Stefels 2009 (Limnol. Oceanogr.: Methods 7, 2009, 595-611) work reporting use of stable isotopes to measure DMSP synthesis rates. Could the authors evaluate their approach to that of Stefels D2O approach – maybe comment whether nanoSIMS can be used in a similar way – as these are both important advances in methodology; somewhere around the last paragraph of the Results and Discussion?*

This is a good point; we have now added a sentence referring to this study in the conclusion. It now reads: “In addition, similarly to other recent stable isotope approaches (Stefels, Dacey and Elzenga, 2009), our method may be used to quantify the production rate of DMSP at the single cell level”.

*It was difficult to follow which images were being referred to, as the individual image files were not labelled. This might be an oddity of the eLife submission system, I accept.*

We are sorry for this issue, we will verify that all figures are labelled prior to re-submission.